# Expression of Hemangioblast Proteins in von Hippel-Lindau Disease Related Tumors

**DOI:** 10.3390/cancers15092551

**Published:** 2023-04-29

**Authors:** Evelynn Vergauwen, Ramses Forsyth, Alexander Vortmeyer, Sven Gläsker

**Affiliations:** 1Department of Neurosurgery, Vrije Universiteit Brussel, Laarbeeklaan 101, 1090 Jette, Belgium; evelynn-v@hotmail.com; 2Department of Neurology, Universiteit Antwerpen, Universiteitsplein 1, 2610 Antwerpen, Belgium; 3Department of Pathology, Universitair Ziekenhuis Brussel, Laarbeeklaan 101, 1090 Jette, Belgium; 4Experimental Pathology Research Group, Vrije Universiteit Brussel, Laarbeeklaan 103, 1090 Jette, Belgium; 5Department of Pathology, Indiana University-Purdue University, 420 University Blvd, Indianapolis, IN 46202, USA; 6Neurosurgery Section, Gesundheitsverbund Landkreis Konstanz (GLKN), 78224 Singen am Hohentwiel, Germany

**Keywords:** von Hippel-Lindau, etiology, pathogenesis, hemangioblast, hemogenic endothelium, Brachyury, TAL1

## Abstract

**Simple Summary:**

Von Hippel-Lindau (VHL) disease is a rare tumor syndrome with autosomal dominant inheritance. We are a team of physicians who established the first VHL disease reference center in Belgium in 2016. Clear cell renal cell carcinomas can be a manifestation of a VHL disease. These tumors are currently a “hot topic” in VHL research, because of the research on the VHL protein pathway by Nobel Prize winner William Kaelin Jr. However, VHL patients also suffer from hemangioblastomas in the cerebellum, retina, and spinal cord. The pathogenesis of these tumors is less well studied. Based on histologic observation of VHL hemangioblastomas, we have gained insights that can be useful for understanding their pathogenesis.

**Abstract:**

Von Hippel-Lindau (VHL) disease is a hereditary tumor syndrome that targets a highly selective subset of organs causing specific types of tumors. The biological basis for this principle of organ selectivity and tumor specificity is not well understood. VHL-associated hemangioblastomas share similar molecular and morphological features with embryonic blood and vascular precursor cells. Therefore, we suggest that VHL hemangioblastomas are derived from developmentally arrested hemangioblastic lineage keeping their potential of further differentiation. Due to these common features, it is of major interest to investigate whether VHL-associated tumors other than hemangioblastoma also share these pathways and molecular features. The expression of hemangioblast proteins has not yet been assessed in other VHL-related tumors. To gain a better understanding of VHL tumorigenesis, the expression of hemangioblastic proteins in different VHL-associated tumors was investigated. The expression of embryonic hemangioblast proteins Brachyury and TAL1 (T-cell acute lymphocytic leukemia protein 1) was assessed by immunohistochemistry staining on 75 VHL-related tumors of 51 patients: 47 hemangioblastomas, 13 clear cell renal cell carcinomas, 8 pheochromocytomas, 5 pancreatic neuroendocrine tumors, and 2 extra-adrenal paragangliomas. Brachyury and TAL1 expression was, respectively, observed in 26% and 93% of cerebellar hemangioblastomas, 55% and 95% of spinal hemangioblastomas, 23% and 92% of clear cell renal cell carcinomas, 38% and 88% of pheochromocytomas, 60% and 100% of pancreatic neuroendocrine tumors, and 50% and 100% of paragangliomas. We concluded that the expression of hemangioblast proteins in different VHL-associated tumors indicates a common embryological origin of these lesions. This may also explain the specific topographic distribution of VHL-associated tumors.

## 1. Introduction

Von Hippel-Lindau (VHL) disease is a rare hereditary tumor syndrome caused by germline mutations of the *VHL* gene on chromosome 3p25.3. Although all cells are affected, VHL disease targets a highly selective subset of organs giving rise to very specific tumors [1,2,3,4]. VHL patients develop central nervous system hemangioblastomas in a remarkable typical topographic distribution throughout the retina, cerebellum, and spinal cord. Other frequent manifestations include renal cysts and clear cell carcinomas, pancreatic cysts, and neuroendocrine tumors, as well as epididymal cysts in males and broad ligament cysts in females. A smaller group of patients is affected by pheochromocytomas, extra-adrenal paragangliomas, and endolymphatic sac or duct tumors [5,6,7,8,9,10]. The biology of organ selectivity and tumor specificity is not well understood. During our research on the pathogenesis of VHL disease, we made three important observations regarding VHL-associated hemangioblastomas. 

First, hemangioblastomas express several embryonic proteins (such as Brachyury and TAL1) and embryonic protein receptors [10,11]. The proteins and receptors mirror the expression profile of an embryonic cell: the hemangioblast. This embryonic cell is a precursor of erythrocytes, vascular smooth muscle cells, and blood vessels. Brachyury and TAL1 are needed for differentiation from mesoderm into hemangioblast and subsequent blood cell precursors.

Second, we observed that hemangioblastomas share similar histomorphological features with the hemangioblast, such as blood island formation and extramedullary hematopoiesis (with VHL-deficient daughter cells) [12]. In addition, many histological similarities exist between different VHL-associated lesions, such as the clear cell aspect.

Third, we discovered preneoplastic lesions in a steady topographic distribution throughout the central nervous system of VHL patients, corresponding to the localization of hemangioblast proteins, such as TAL1, during embryogenesis [13]. 

Based on these three observations, we have hypothesized that VHL-associated hemangioblastomas originate from developmentally arrested embryonic hemangioblasts containing a neoplastic potential and the protracted capacity to differentiate [12,14]. 

The presence of embryonic hemangioblast proteins (Brachyury and TAL1) has not yet been investigated in VHL-associated tumors other than hemangioblastomas. Due to the common morphological features and the common molecular basis of the different VHL tumors, it is of major interest to investigate whether VHL-associated tumors, other than hemangioblastomas, share these aforementioned features. 

## 2. Materials and Methods

### 2.1. Tissue Selection

The study protocol was approved by the local research ethics service (reference number 2016/271). To access surgically resected tumors of VHL patients, a query through the Belgian Cancer Registry (national central tumor databank) was performed. Additionally, colleague neurosurgeons throughout Belgium were contacted, and several anatomic pathological institutions were visited. Specimens were collected from the following institutions: University Hospital Brussels (Jette, Belgium), Jules Bordet Institute (Brussels, Belgium), Erasme University Hospital Brussels (Anderlecht, Belgium), University Hospital Saint-Luc (Woluwe-Saint-Lambert, Belgium), University Hospital Liège (Liège, Belgium), and University Hospital Antwerp (Edegem, Belgium). Other samples were contributed by the biomaterial database of the German VHL family alliance (Verein VHL betroffener Familien e.V.) at the Dr. Margarete Fischer-Bosch-Institut für Klinische Pharmakologie (IKP) at the Robert-Bosch-Krankenhaus (RBK) (Stuttgart, Germany). 

Any tumor related to VHL disease was included in the samples cohort (central nervous system hemangioblastomas, clear cell renal cell carcinomas, pheochromocytomas, pancreatic tumors, endolymphatic sac tumors, epididymis/broad ligament cystadenomas and extra-adrenal paragangliomas). Patients of all ages and both genders, with either a genetical or clinical confirmation of VHL disease were included.

Seventy-five surgically resected tumors from 51 different von Hippel-Lindau patients were obtained (27 cerebellar hemangioblastomas, 20 spinal hemangioblastomas, 13 clear cell renal cell carcinomas, 8 pheochromocytomas, 5 pancreatic tumors, and 2 extra-adrenal paragangliomas). In total, 20 male and 31 female patients were included, their ages ranging from 13 to 70 years old. All patients previously had either genetical or clinical confirmation of VHL disease. Patients had been operated on between 1989 and 2018. Patient characteristics and tumor data are summarized in Table 1. 

### 2.2. Tissue Preparation and Histological Classification

In-house tumors and control tissues were fixed in 10% neutral-buffered formalin and embedded in paraffin. Tumors obtained from other institutions had been processed in different devices, however, with the same formalin concentration. Tumors and control tissue blocks were cut in 4 µm sections, mounted on TOMO slides (Matsunami Glass IND Ltd., Osaka, Japan) and dried for 1 h at 50 °C. Haematoxylin and eosin (H & E) staining was performed on all sections for diagnostic confirmation of tumor type. Clear cell renal cell carcinomas, pheochromocytomas, pancreatic neuroendocrine tumors, and extra-adrenal paragangliomas were histologically graded according to the following grading systems: World Health Organization (WHO)/International Society of Urological Pathology (ISUP) grading system [15], Pheochromocytoma of the Adrenal Gland Scaled Score (PASS) [16], WHO grading system [17], and for paragangliomas again the Pheochromocytoma of the Adrenal Gland Scaled Score (PASS) [16]. 

### 2.3. Immunohistochemistry

All experiments were performed in the laboratory of anatomic pathology at the University Hospital Brussels. Tumor and control sections were stained in the same way for each run, using an automated slide staining system (BenchMark ULTRA, Ventana Medical Systems, Inc., Oro Valley, AZ, USA) with standard settings. All used products were produced by Ventana Medical Systems, Inc, Oro Valley, Arizona, United States. In brief, sections were manually preheated for 20 min at 60 °C, then heated automatically for 4 min at 72 °C and deparaffinized with EZ Prep. Next, Long Cell Conditioner 1 and CC Medium Coverslip were alternately applied at 95 °C for breaking of formalin bonds/antigen retrieval and rehydration, respectively. Sections were rinsed with a Tris based Reaction Buffer at 36 °C. ISH Peroxidase Inhibitor was applied to decrease non-specific background staining, and sections were rinsed with several cycles of Tris based Reaction Buffer wash at 36 °C. Secondary antibodies were diluted in Antibody Diluent and applied manually. HQ Universal Linker, HRP Multimer, and Amplifier H_2_O_2_ (OptiView Amplification Kit) were applied for optimal linking of primary to secondary antibody, alternated by rinsing cycles of Tris based Reaction Buffer. Incubation with the secondary antibody was performed for 8 min (anti-TAL1) and 2 h (anti-Brachyury) at 36 °C. The tertiary antibody was visualized with hydrogen peroxide substrate and three, 3′-diaminobenzidine tetrahydrochloride (DAB) chromogen (ultraView Universal DAB Detection Kit). Sections were counterstained with Mayer’s haematoxylin, postcounterstained with bluing agent, manually dehydrated with 70–100% graded alcohol series and xylene, and cover-slipped. 

Secondary antibodies were validated on validated positive and negative (tissue microarray) control tissues in our institution. Data about concentration, dilution, distributor, cellular localization, and positive control tissues are summarized in Table 2. Human chordoma tumor was used as positive control tissue for Brachyury. A human T-cell acute lymphoblastic leukemia cell line was grown and used as positive control tissue for TAL1. Tissue microarray was used as a negative control, containing different human tissues (spleen, stomach, appendix, colon, liver, pancreas, kidney, adrenal gland, testis, prostate, lung, tonsil, cerebrum, cerebellum, and spinal cord). Brachyury and TAL1 sporadically stained isolated cells in the negative control tissues (presumably mast cells).

### 2.4. Evaluation of Immunohistochemical Staining

To date, there are no standard immunohistochemical (IHC) scoring systems for the antibodies that were used [18]. Different semi-quantitative IHC scoring systems were evaluated. It was concluded that the Remelle composite scoring system (immunoreactive score, IRS) suits our study best [19]. IHC staining was scored as intensity of staining signal multiplied by percentage of positive tumor cells. Intensity was classified as 0 (no detectable staining), 1 (weak, visible at high to intermediate magnification: ×40; nuclear: faint, membranous: only part of the membrane), 2 (moderate, visible at intermediate to low magnification: ×10–20; nuclear: moderate, membranous: large part), and 3 (strong, visible at low magnification: ×2.5–5; nuclear: strong, membranous: strong, whole membrane, and also: same intensity as control tissue). Percentage of positive tumor cells was evaluated as 0 (0%), 1 (<10%), 2 (10–50%), 3 (51–80%), or 4 (>80%). Scores for different VHL-related tumors were given as mean values +/− SD. Ultimately, tumor and control tissues were classified according to ‘no expression’ (total score 0), ‘low expression’ (total score 1 to 4) or high ‘expression’ (total score 5 to 9). 

### 2.5. Statistical Analysis

All statistical data analyses were conducted with Real Statistics Resource Pack for Excel 2016 (Microsoft Office, Microsoft, Redmont, Washington, DC, USA). Prevalence of embryonic marker expression in different types of VHL tumors was summarized using descriptive statistics. The nonparametric Kruskal Wallis test was used to determine if there was a statistical difference in embryonic marker expression—represented by the Remelle composite scoring system (dependent variable) between VHL tumors (independent variable). The differences were reported in medians. 

## 3. Results

### 3.1. Hemangioblastomas

All 47 hemangioblastomas (27 cerebellar, 20 spinal) corresponded histologically to WHO grade I and consisted of large round, oval, or polygonal tumor cells in a capillary vascular network. In 13 of 47 hemangioblastomas (3 cerebellar, 10 spinal), moderate to strong nuclear Brachyury expression was seen; mostly in half of the tumor cell population. Two of 20 spinal hemangioblastomas showed strong nuclear Brachyury staining in almost all tumor cells (an example can be seen in Figure 1). Weak to moderately strong cytoplasmic Brachyury expression was seen in 8 of 47 hemangioblastomas (5 cerebellar, 3 spinal), covering various amounts of tumor areas, including three tumors with nuclear expression as well. Figure 1 shows an example of cytoplasmic Brachyury staining in a cerebellar hemangioblastoma. 

Sometimes weak, but mostly moderately strong, nuclear TAL1 expression was seen in 43 of 47 hemangioblastomas (25 cerebellar, 18 spinal), staining approximately half of the tumor area. TAL1 cytoplasmic expression was seen in 40 of 47 hemangioblastomas (22 cerebellar, 18 spinal), covering more than half of the tumor area. 

Brachyury and/or TAL1 staining was never seen in the whole tumor cell population. Tumors that expressed Brachyury always co-expressed TAL1, but not vice versa. Tumor cells surrounding cysts had the same staining pattern. Sometimes, weak to strong cytoplasmic Brachyury and TAL1 staining was seen in some, but not all, endothelial cells that were contained by positive nuclear or cytoplasmic tumor areas, and sometimes in negative areas. Data are summarized in Appendix A.

### 3.2. Clear Cell Renal Cell Carcinomas

Two of 13 clear cell renal cell carcinomas showed weak to moderate nuclear Brachyury staining, more specifically in a small part of the tumor cell population. Weak Brachyury cytoplasmic staining was seen in three tumors, in less than half of the tumor area, arranged in small groups or cyst walls. Additionally, one of these cytoplasmic-staining tumors contained few tumoral and nearby endothelial cells with strong nuclear Brachyury expression (<10% of tumor cells). Twelve of 13 renal cell carcinomas showed moderate to strong nuclear TAL1 expression, varying from small to mostly very large areas (see Figure 2). Eleven renal cell carcinomas showed a weak to moderate cytoplasmic TAL1 expression in large tumor areas. Occasionally, TAL1 was weakly expressed in endothelial cytoplasm as well. Similar to hemangioblastomas, tumors with nuclear Brachyury expression always co-expressed TAL1, but not vice versa. Data are summarized in Appendix A.

### 3.3. Pheochromocytomas

In three of eight pheochromocytomas, weak to moderately strong cytoplasmic Brachyury expression was found in groups of tumor cells, or the whole population in one tumor (an example can be seen in Figure 2). Nuclear Brachyury expression was not seen, except in isolated sporadic cells, in all tumors. Moderate to strong nuclear TAL1 staining was seen in three pheochromocytomas in a small part of the tumor area. Weak to moderate cytoplasmic TAL1 staining was seen in seven of eight pheochromocytomas, mostly in large areas. Some endothelial cells showed weak and incomplete cytoplasmic Brachyury staining, however, in one pheochromocytoma a vascular structure was found with moderate Brachyury expression in nuclei of plump endothelial cells. TAL1 was sometimes weakly expressed in endothelial cytoplasm as well. Data are summarized in Appendix A.

### 3.4. Pancreatic Neuroendocrine Tumors

Two of five pancreatic neuroendocrine tumors showed strong nuclear Brachyury expression in a small population of tumor cells. In between tumor cell groups, endothelial cells showed weak to moderate cytoplasmic Brachyury staining. Another tumor showed weak cytoplasmic Brachyury staining in a small population of tumor cells. Normal pancreatic tissue showed negligible cytoplasmic staining with Brachyury and was negative for TAL1.

In three of five tumors, moderately strong nuclear TAL1 staining was observed in a larger population of tumor cells. Some endothelial cells showed cytoplasmic staining with TAL1 (but not in all tumors), usually in larger vessels with plump endothelial cells, and sometimes large vascular structures surrounded by tumor cells. Weak to moderate cytoplasmic TAL1 staining was seen in large areas of four tumors (see Figure 2). In comparison to other VHL tumors, Brachyury and TAL1 staining was more dissociated. Data are summarized in Appendix A.

### 3.5. Extra-Adrenal Aragangliomas

In one of two paragangliomas, small tumor areas showed strong nuclear and weak cytoplasmic Brachyury expression. In addition, this paraganglioma showed strong nuclear and moderate cytoplasmic TAL1 expression in a large area (Figure 2). In the other paraganglioma, only moderately strong cytoplasmic expression was seen in almost all tumor cells. In both tumors, some vascular structures showed weak cytoplasmic Brachyury staining. Data are summarized in Appendix A.

### 3.6. Statistical Analysis

Brachyury and TAL1 expression was, respectively, observed in 26% and 93% of cerebellar hemangioblastomas, 55% and 95% of spinal hemangioblastomas, 23% and 92% of clear cell renal cell carcinomas, 38% and 88% of pheochromocytomas, 60% and 100% of pancreatic neuroendocrine tumors, and 50% and 100% of paragangliomas. Median and mean Remelle scores are described in the Appendix A. The difference between Remelle scores (both nuclear and cytoplasmic) was statistically significant between different VHL tumors. In general, TAL1 staining was stronger compared to Brachyury staining in all VHL tumors (according to Remelle scores). Additionally, there was a statistically significant difference in amount of staining between different tumors, which could be ranked according to decreasing nuclear Brachyury expression: spinal hemangioblastomas, extra-adrenal paragangliomas, pancreatic neuroendocrine tumors, cerebellar hemangioblastomas, clear cell renal cell carcinomas, and pheochromocytomas. Cytoplasmic Brachyury staining was mostly observed in pheochromocytomas, cerebellar hemangioblastomas, and spinal hemangioblastomas. 

Ranking for amount of TAL1 staining was as follows: cerebellar hemangioblastomas, clear cell renal cell carcinomas, spinal hemangioblastomas, extra-adrenal paragangliomas, pancreatic neuroendocrine tumors, and pheochromocytomas. Cytoplasmic TAL1 staining was observed in all VHL tumors and was strongest in extra-adrenal paragangliomas, cerebellar hemangioblastomas, and spinal hemangioblastomas.

## 4. Discussion

VHL disease targets a highly selective subset of organs with very specific types of tumors. To date, this principle of organ selectivity and tumor specificity, as well as the origin of the “stromal tumor cell” of VHL tumors, has been an ongoing subject of debate, and its growth mechanisms remain largely unknown. 

Per this study, evidence for the expression of hemangioblast proteins in different types of VHL-associated tumors is provided, including cerebellar and spinal hemangioblastomas, clear cell renal cell carcinomas, pheochromocytomas, pancreatic neuroendocrine tumors, and paragangliomas. The expression of hemangioblast proteins in different types of VHL tumors provides evidence to the pre-hemangioblastic stem cells as a key role player in the origin of the different VHL-associated tumors. This point of view could be a possible explanation for the principle of organ selectivity and tumor specificity in VHL disease. We noted with interest that the distribution of TAL1 expression in mammalian embryonic development matches the vast topographic distribution of VHL tumors [20] (Figure 3).

### 4.1. The Embryonic Hemangioblast: Fact or Fiction?

The irrefutable existence of the embryonic hemangioblast in vivo has not been confirmed to date. In 1880, Wilhelm His was the first to hypothesize a common ancestor cell for hematopoietic cells and endothelium [21], a concept which was consolidated in 1917 by Sabin Florence on living chick embryos. Sabin named the progenitor cell “the angioblast” [22]. Murray P.D.F. later renamed the ancestor cell as “the hemangioblast” [23]. 

Beyond question, hematopoietic and endothelial cell lineages share common typical mesodermal markers (such as Brachyury, BMP4, and VEGFR2), surface markers (such as CD34, VE cadherin, CD31), and transcription factors (such as RUNX1 and GATA2) throughout their development. Nevertheless, two different hypotheses co-exist in the current literature. Some researchers have hypothesized that endothelial cells and hematopoietic stem/progenitor cells derive from independent epiblast populations. The second hypothesis states that the mesodermally-derived bipotent hemangioblast gives rise to endothelial cells and hematopoietic stem/progenitor cells, whether or not after a transitional phase of hemogenic endothelium [24,25,26,27]. 

### 4.2. Hypothesis of Embryonic Origin 

The multiplicity, histological heterogeneity, and specific topographic distribution of angiomatous tumors in “von Hippel’s disease” or “angiomatosis retinae” made Arvid Lindau (1926) originally hypothesize that “cerebellar (capillary hem) angiomas”, being part of “central nervous system angiomatosis”, originated from a vascular congenital maldevelopment. The specific region of interest was a mesoderm-derived capillary network in the inferior medullary velum, its malformation taking place in the third fetal month [28]. This vascular network was described by Karlefors [29]. Based on the observation of immature vascular elements and blood cell formation, Lindau, Cushing, and Bailey (1928), and Stein (1960), hypothesized that the original cells of “(hem) angioblastomas” were maldeveloped angioblastic cells which, later on, transformed into fat-laden “pseudoxanthoma tumor cells”: now known as the intravascular tumor stromal cell. They all described the simultaneous occurrence of mature and immature vascular elements in cerebellar hemangioblastomas [8,9,30,31]. 

Notwithstanding its ingenuity, the embryonic hypothesis was never proven on all DNA, RNA, and protein levels, and it does not completely cover the broad neoplastic and extra-nervous system diversity of the VHL syndrome. Since 1926, many other origin cells have been suspected and conflicting immunohistochemical expression patterns have emerged ever since, as summarized by Ma and Zu [32]. 

Not until recently, the embryonic hypothesis regained attention by the discovery of early embryonal maldevelopmental structures: developmentally arrested structural elements (DASEs). They have been found in supposedly normal cerebellum and dorsal nerve roots on surgical and autopsy tissue from VHL patients. Interestingly, DASEs already exhibit loss of *VHL*-heterozygosity (in contrast to normal surrounding tissue), and some have the ability to transform into frank VHL hemangioblastomas with mesenchymal components. Indeed, the more progressive epithelioid hemangioblastomas feature VHL-deficient vascular structures and extramedullary haematopoiesis, which may also explain tumor diversity in VHL disease [13,33,34,35,36,37,38]. Comparable embryonal precursor tissue was found in the kidney, epididymal and endolymphatic duct and sac from VHL patients [14].

One major finding of this study is that stronger evidence is provided for the embryonic hypothesis by proposing the hemangioblast as the cell of origin of a broader spectrum of VHL tumors. It was shown that the cell’s typical proteins Brachyury and TAL1 are expressed by a broader spectrum of VHL tumors rather than by hemangioblastomas only. 

Our previous research showed the expression of Brachyury and TAL1 in 10 of 10 VHL-related hemangioblastomas [39]. Ma et al. and Park et al. later confirmed the expression of Brachyury and TAL1 in another small series of VHL-related hemangioblastomas, of which the exact amount of positively staining tumors was not mentioned [32,40,41]. In our current study some hemangioblastomas showed very specific nuclear and/or cytoplasmic Brachyury expression in select tumor areas, with varying staining intensities, suggesting different tumor cell development stages. 

### 4.3. New Insights into Brachyury Distribution

Interestingly, this manuscript is the first that describes both nuclear and cytoplasmic Brachyury expression in VHL hemangioblastoma tumor cells. Thus far, two articles described exclusive cytoplasmic expression in 20 of 22 non-VHL hemangioblastomas [42,43], reserving nuclear expression for human chordomas. In some chordomas, also cytoplasmic can be observed although in lower dilutions [44,45,46,47]. Tirabosco et al. described nuclear expression in both intra- and -extra axial bony chordomas and hemangioblastomas, without information about *VHL*-status. They are also the first to report Brachyury expression at the protein level in normal adult tissue: the testis [47]. An article from Doyle et al. describes the absence of either nuclear and cytoplasmic Brachyury expression in 22 peripheral VHL and non-VHL hemangioblastomas [48]. These discrepancies might be due to different Brachyury isoforms of different antibodies, as previously suggested by Hamilton et al. [49].

## 5. Conclusions

Based on the abovementioned observations and results, it is hypothesized that VHL tumors originate from the pre-hemangioblastic lineage, a hemangioblast daughter cell (also known as “hemogenic endothelium”) or that they have at least a hemangioblast-like-phenotype. Additionally, this hypothesis (especially with regard to TAL1) may explain the vast topographic distribution of VHL tumors in a highly selective subset of organs.

## Figures and Tables

**Figure 1 cancers-15-02551-f001:**
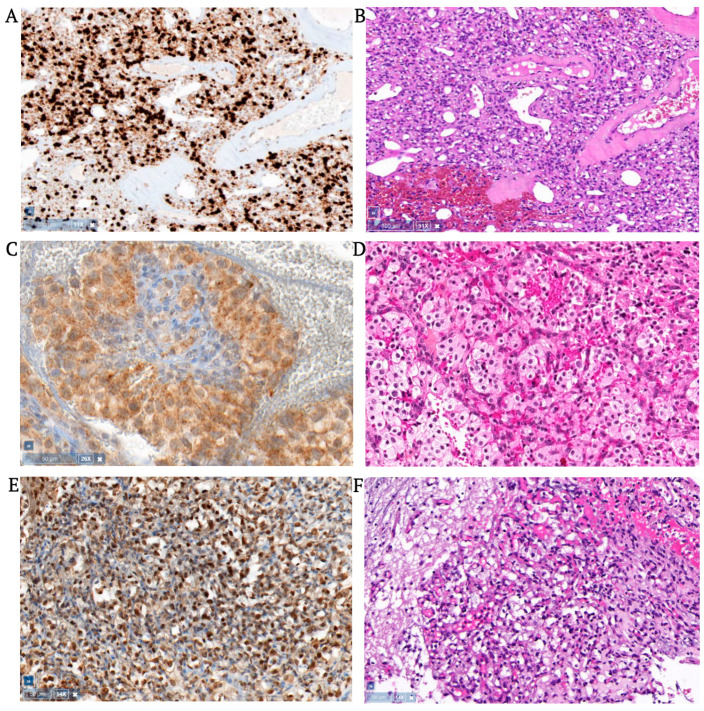
Expression of Brachyury and TAL1 in VHL hemangioblastomas. Panel (**A**) shows spinal hemangioblastoma with nuclear Brachyury staining with Remelle score 12 (i.e., strong staining in >80% of tumor cells), and (**B**) H & E staining. Panel (**C**) shows cerebellar hemangioblastoma with cytoplasmic Brachyury staining with Remelle score 8 (i.e., moderate staining in >80% of tumor cells), and (**D**) H&E staining. Panel (**E**) shows cerebellar hemangioblastoma with nuclear TAL1 staining with Remelle score 9 (i.e., strong staining in 51–80% of tumor cells), and (**F**) H & E staining. (Screenshot made by Pathomation Digital Pathology Software v2.1.1.1985).

**Figure 2 cancers-15-02551-f002:**
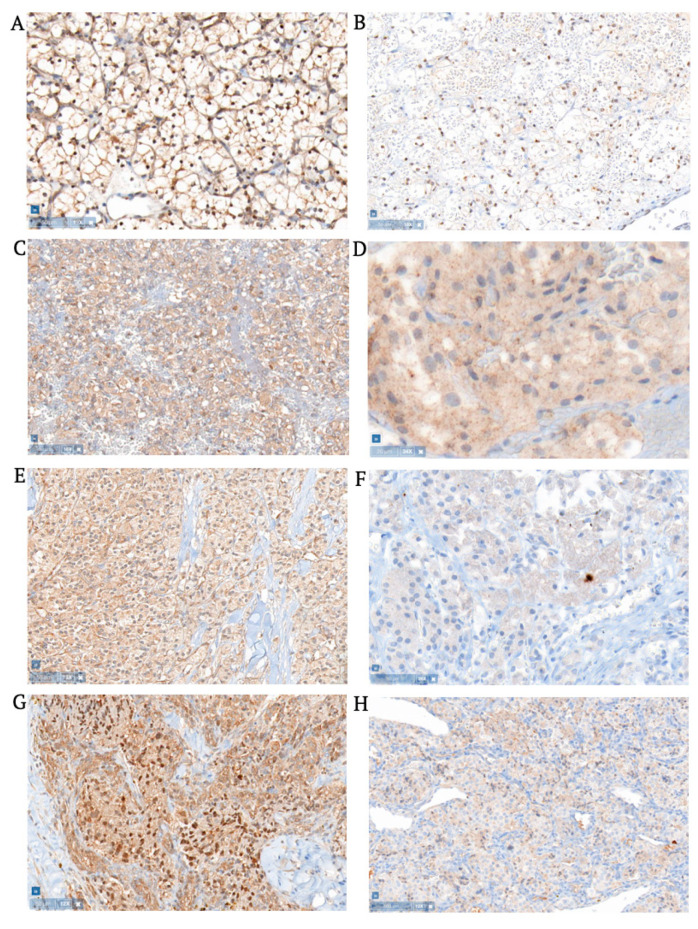
Expression of TAL1 and Brachyury in other VHL-related tumors. Clear cell carcinoma with (**A**) nuclear TAL1 staining with Remelle score 12 (i.e., strong staining in >80% of tumor cells), and (**B**) nuclear Brachyury staining with Remelle score 2 (i.e., weak staining in 10–50% of tumor cells). Pheochromocytoma with (**C**) cytoplasmic TAL1 staining with Remelle score 4 (i.e., moderate staining in 10–50% of tumor cells), and (**D**) cytoplasmic Brachyury staining with Remelle score 4 (i.e., moderate staining in 10–50% of tumor cells). Pancreatic neuroendocrine tumor with (**E**) cytoplasmic TAL1 staining with Remelle score 6 (i.e., moderate staining in 51–80% of tumor cells), and (**F**) cytoplasmic Brachyury staining with Remelle score 2 (i.e., weak staining in 10–50% of tumor cells). Extra-adrenal paraganglioma with (**G**) cytoplasmic and nuclear TAL1 staining with Remelle score 8 (i.e., moderate staining in >80% of tumor cells), and (**H**) cytoplasmic Brachyury staining with Remelle score 4 (i.e., moderate staining in 10–50% of tumor cells). (Screenshots made by Pathomation Digital Pathology Software v2.1.1.1985).

**Figure 3 cancers-15-02551-f003:**
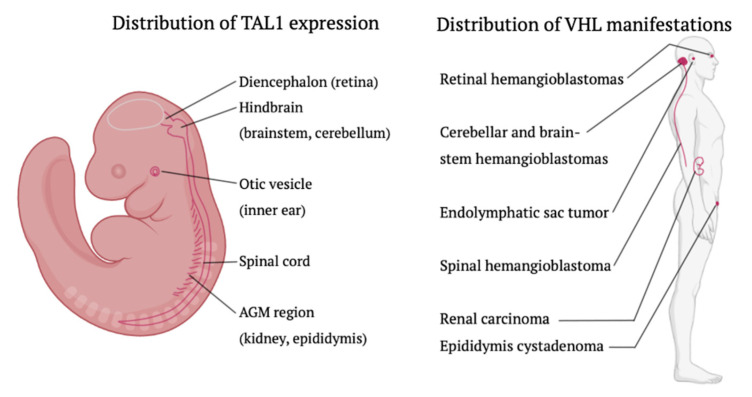
The distribution of TAL expression during normal embryonic development closely recapitulates the distribution of VHL manifestations. Note that for every anatomic location of intraembryonic TAL1 expression (**left**), there is a corresponding VHL manifestation (**right**). Image design with Biorender and SketchBook.

**Table 1 cancers-15-02551-t001:** Clinical and pathological findings.

Case	Sex	Age (Year)	Tumor Site	Grade	Tumor Size (mm³)	Remelle Brachyury		Remelle TAL1	
						Nuclear	Cytoplasmic	Nuclear	Cytoplasmic
Cerebellar hemangioblastoma				WHO					
1	M (18)	17	unknown	1	unknown	3	0	2	3
2	M (6)	17	right hemisphere	1	2300	0	0	9	0
3	M (6)	18	left hemisphere	1	4200	0	0	6	4
4	M (6)	20	left hemisphere	1	48,000	0	0	9	6
5	M (18)	25	right hemisphere	1	9880	0	0	4	6
6	F (7)	26	unknown	1	3491	0	0	0	0
7	M (3)	27	unknown	1	1280	3	0	4	8
8	F	30	left hemisphere	1	356	0	1	9	6
9	M (2)	30	left hemisphere	1	700	0	0	4	6
10	F (7)	30	unknown	1	unknown	0	0	6	6
11	M (1)	31	right hemisphere	1	1601 (3 tumors)	0	0	2	6
12	F	32	right hemisphere	1	2265 (2 tumors)	0	0	6	6
13	M (1)	37	midline	1	1760 (2 tumors)	0	0	6	6
14	M (4)	41	left hemisphere	1	4752	0	0	6	6
15	F (8)	42	left hemisphere	1	unknown	0	0	4	6
16	M (1)	43	left + right hemisphere	1	7560 (2 tumors)	0	8	9	6
17	F (5)	43	intraventricular	1	1920	0	0	6	6
18	F (8)	47	right hemisphere	1	288	0	3	6	6
19	M (4)	50	unknown	1	9478 (2 tumors)	3	2	4	6
20	F	55	left hemisphere	1	4756 (2 tumors)	0	0	6	6
21	M	56	right hemisphere	1	4048	0	0	6	6
22	F	65	right hemisphere	1	9525	0	3	12	6
23	F	67	vermis	1	1728	0	0	0	6
24	F	22	right hemisphere	1	5400	0	0	4	6
25	M	44	left hemisphere	1	532	0	0	4	6
26	F (19)	40	right hemisphere	1	1241	0	0	6	6
Spinal hemangioblastoma				WHO					
27	M (6)	17	C6	1	1000	0	0	4	2
28	F	24	C1 + C5	1	92 (2 tumors)	0	0	2	4
29	F	24	T7–8	1	192	3	0	6	0
30	M (15) *	25	medulla oblongata	1	unknown	0	0	0	2
31	F (9)	26	medulla, obex	1	3588	6	0	6	6
32	F (10)	26	T10–11	1	250	0	0	12	8
33	F (9)	27	C5–6	1	224	0	2	9	6
34	F (10)	27	C5–6	1	6	9	0	2	6
35	M (3)	30	T12-L1	1	3360	12	0	1	3
36	M (14)	31	nerve root	1	1500	6	0	6	6
37	F	34	T5	1	1080 (2 tumors)	0	0	9	4
38	F (11)	34	T10	1	1000	2	0	4	2
39	F	36	C3-C5	1	6	3	0	2	1
40	M	38	C1-C3	1	3600	3	6	6	6
41	F	42	C4-C6	1	180	3	3	4	6
42	F (12)	42	lumbar nerve root	1	1728	6	0	4	2
43	M (13)	47	filum terminale	1	1600	0	0	6	6
44	F	58	T12	1	43	0	0	0	0
45	F	33	borderzone cerebellum	1	225	0	0	6	4
46	F (19)	37	C4–C5	1	91	0	0	3	4
47	M	70	L1	1	2856	0	0	4	8
Clear cell renal cell carcinoma				WHO/ISUP					
48	F (11)	31	left kidney	2	23,400	0	1	0	0
49	F (16)	37	right kidney	1	4335	0	0	2	0
50	M (14)	37	right kidney	2	87,500	0	0	9	6
51	M	38	right kidney	1	5832	0	0	2	0
52	F (16)	38	left kidney	1	unknown	2	1	2	0
53	F	46	left kidney	1	389,017	0	0	12	0
54	F (12)	46	left kidney	1	6000	0	0	6	3
55	M (13)	47	right kidney	1	7820	0	0	12	6
56	F (5)	52	right kidney	1	8000	2	2	4	4
57	M (17) *	52	right kidney	2	2197	0	0	2	8
58	F	57	unknown	1	12,167	0	0	12	0
59	F	35	right kidney	2	10,166	0	0	6	8
60	M	44	left kidney	2	123,786 (11 tumors)	0	0	6	4
Pheochromocytoma				PASS (Tompson)					
61	M (2)	13	right adrenal gland	6	192,500	0	0	0	1
62	M	20	right adrenal gland	0	2240	0	0	4	4
63	F (11)	21	right adrenal gland	3	7920	0	2	0	0
64	M (6)	24	right adrenal gland	3	210	0	0	0	1
65	M (15) *	46	right adrenal gland	0	2520	0	0	3	4
66	M (17) *	47	right adrenal gland	2	14,520	0	0	0	8
67	M	51	left adrenal gland	0	21,000	0	2	2	4
68	F	54	unknown	7	61,845	0	4	0	8
Pancreatic neuroendocrine tumor				WHO/AJCC					
69	F	23	pancreatic head	2	10,648	0	0	2	6
70	F	29	pancreatic head	4	8800	0	0	4	0
71	F	45	pancreatic head	3	unknown	3	0	0	4
72	F	46	pancreatic head + body	3	unknown	0	1	0	2
73	M	48	pancreatic head	3	unknown	3	0	2	4
Extra-adrenal paraganglioma									
74	F	31	carotid body	0	20,825	0	0	0	8
75	M	60	carotid body	1	151,740	3	1	6	8

M = male, F = female, * = twin brothers, numbers between brackets = same patient. WHO: World Health Organization; ISUP: International Society of Urological Pathology; PASS score: Pheochromocytoma of the Adrenal gland Score; AJCC: American Joint Committee on Cancer.

**Table 2 cancers-15-02551-t002:** Data about antibodies.

Target Protein	Type of Antibody	Concentration, Dilution	Distributor	Cellular Localization	Positive Control
TAL1	Rabbit polyclonal IgG to Tal1 C-terminal (ab155195)	1 mg/mL, 1:2000	Abcam, Cambridge, UK	Nuclear/cytoplasmic	Human T-ALL (DSMZ)
Brachyury	Rabbit monoclonal IgG [EPR18113] to Brachyury (ab209665)	0.413 mg/mL, 1:50	Abcam, Cambridge, UK	Nuclear/cytoplasmic	Human chordoma

## Data Availability

Data is contained within the article or Appendix A.

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
