# Peer review of "Expression of Hemangioblast Proteins in von Hippel-Lindau Disease Related Tumors"

_cancers, 2023, doi:10.3390/cancers15092551_

Round 1

Reviewer 1 Report

In this article, authors describe expressions of hemangioblast proteins in VHL related tumors. This manuscipt is well written, however, the following critical points should be revised.

1. Figure 1 shows expression of Brachyury in VHL hemangioblastoma. Hemangioblastomas reveal nuclear or cytoplasmic staining. Authos sholud explain why Brachyury is localized at both nucleus and cytoplasm in VHL hemangioblatoma.  Also, a microphotograph of figure 1 (b)  dose not appeared to be CNS hemangioblastoma because nuclei of tumor cells are large authough nucleus of CNS hemangioblastoma is usually small. Authors should provide HE staining of  figure 1 (b). Furthermore, ratio of immunoreactibe cells in figure 1 would be descibed. 

2. Figure 2 shows Brachyury and TAL in VHL related tumors. However, all VHL related tumors are examined for only either Brachyury or TAL. Authors should show immunostainings of both Brachyury and TAL1 in all VHL related tumors. 

3. Figure 3 is almost same as figure 3 of the paper: Glasker S, et al. Hemangioblastomas share protein expression with embyonal hemangioblast progenitor cells.  Cancer Res 66: 4167-4172, 2006. Authors should exchange from it to an original fugure. 

Reviewer 2 Report

This study examines the expression of hemangioblast proteins brachyury and TAL1 to provide an explanation to a longstanding question on why VHL is the only hereditary cancer syndrome that results in hemangioblastomas.

This manuscript is a nicely designed pathology-based study, which uses VHL patient tissue to validate the initial hypothesis.  It is clear and well laid out.

My only minor comment would be if any VHL genotype information is available that should be added.  And if there are any correlations of VHL subtypes (type 1, type 2a etc).

Author Response

This study examines the expression of hemangioblast proteins brachyury and TAL1 to provide an explanation to a longstanding question on why VHL is the only hereditary cancer syndrome that results in hemangioblastomas. This manuscript is a nicely designed pathology-based study, which uses VHL patient tissue to validate the initial hypothesis.  It is clear and well laid out. My only minor comment would be if any VHL genotype information is available that should be added.  And if there are any correlations of VHL subtypes (type 1, type 2a etc).

Thank you for this worthwhile and constructive comment. Unfortunately, we only have genotype information of the patients with cerebellar hemangioblastomas and of some patients with spinal hemangioblastomas, because these were the patients who had been operated by the corresponding author (S.G.). The other patients had been operated in different institutions and the clinico-pathological reports only mentioned “genetically confirmed case”. We chose not to include this incomplete information.

Round 2

Reviewer 1 Report

In this article, authors describe "Expression of hemangioblast proteins in VHL related tumors".  This manuscript is well improved and  is highly evaluated.